# Mu-Net a Light Architecture for Small Dataset Segmentation of Brain Organoid Bright-Field Images

**DOI:** 10.3390/biomedicines11102687

**Published:** 2023-09-30

**Authors:** Clara Brémond Martin, Camille Simon Chane, Cédric Clouchoux, Aymeric Histace

**Affiliations:** 1ETIS Laboratory UMR 8051 (CY Cergy Paris Université, ENSEA, CNRS), 6 Avenue du Ponceau, 95000 Cergy, France; 2Witsee, 33 Ave. des Champs-Élysées, 75008 Paris, France

**Keywords:** brain organoid, segmentation, U-Net, Mini-UNet, MU-Net, bright-field

## Abstract

To characterize the growth of brain organoids (BOs), cultures that replicate some early physiological or pathological developments of the human brain are usually manually extracted. Due to their novelty, only small datasets of these images are available, but segmenting the organoid shape automatically with deep learning (DL) tools requires a larger number of images. Light U-Net segmentation architectures, which reduce the training time while increasing the sensitivity under small input datasets, have recently emerged. We further reduce the U-Net architecture and compare the proposed architecture (MU-Net) with U-Net and UNet-Mini on bright-field images of BOs using several data augmentation strategies. In each case, we perform leave-one-out cross-validation on 40 original and 40 synthesized images with an optimized adversarial autoencoder (AAE) or on 40 transformed images. The best results are achieved with U-Net segmentation trained on optimized augmentation. However, our novel method, MU-Net, is more robust: it achieves nearly as accurate segmentation results regardless of the dataset used for training (various AAEs or a transformation augmentation). In this study, we confirm that small datasets of BOs can be segmented with a light U-Net method almost as accurately as with the original method.

## 1. Introduction

Brain organoids (BOs) are 3D in vitro brain-like cultures that are paving the way for promising alternatives to in vivo models for analyzing brain structure and function and performing personal diagnosis for drug screening. Indeed, these cultures replicate cell type complexity (for instance, communication), the self-organization of tissues, and brain organization: they present various distinct cerebral regions, such as ventricular zones [1]. However, BOs suffer from batch syndrome: BOs in the same culture environment do not necessarily develop the same morphology [2]. To individually follow the development of cultures that model physiological or pathological development, neurobiologists use image analysis tools that are often not automated and adapted to the nonspherical shape of the cultures [3].

In 2020, a team developed software to analyze the fluorescence volumes of intact BOs [4]. This software includes a proposed image segmentation tool that extracts the object shape from its background using the well-known U-Net architecture to characterize the morphology of ventricular zones with transformation data augmentation strategies (flip-flop, rotation) [4]. U-Net is an encoder–decoder network created by [5] to segment annotated images [5]. The contracting path captures the context of images and enables cell localization. In a previous work, we proposed using U-Net with various data augmentation optimization strategies in an adversarial autoencoder (AAE) (which synthesizes natural and variable images from the original ones) during the training step of the segmentation process to extract the shape of BOs in bright-field images [6]. However, this architecture does not seem suited for BO extraction due to some issues: the prolonged training time, the image number requirements (and classic data augmentation strategies are insufficient for reaching a high enough accuracy), and poor neuroepithelial zone segmentation accuracy depending on the optimization used (essential for determining whether the brain culture is growing well after 13 days of culture).

Since its first implementation, U-Net has become a reference for biomedical object segmentation, and recent developments could help progress brain organoid segmentation [7]. For instance, HookNet includes a multibranch encoder–decoder approach, with one branch used for contextual information and the other for target information [8], and BB-Unet, a reduction layer and a bounding box prior filtering approach [9]. Ref. [10] reduces only convolutional layers and uses dropout regularization [10]. In U-Net++, the structure is lightened by employing densely nested strip pathways [11]. The stripped-down U-Net (SD-UNet) is composed of depthwise separable convolutions with a weighted standardization to accelerate performance and decrease computing time. This model contains fewer parameters to learn, a smaller architecture and a shorter computation time than U-Net [12]. In contrast to SD-Unet, SmaAT-UNet adds an attention module alongside use of depthwise separable convolutions [13]. AFF-UNet uses skip connections with an attention channel and a spatial attention module [14]. DMF-AU includes multiscale constraints on the backbone to reduce information loss and lighten the architecture for small datasets [15]. This network employs a differential match filtering layer to identify structures and anisotropic attention to reinforce information on the most representative features. Others use cascaded networks to reduce the number of parameters to learn [16]. In the CNL-UNet structure, a pretrained encoder is proposed to allow the segmentation of small datasets. By using the transfer learning properties of the encoder, the modified skip connections for reducing gaps between mirrored layers and the classifier and localizer module CNL, they efficiently learn the small amount of data [17]. Another encoder architecture update consists of using transformers after convolutional layers to lighten the architecture, such as in LeViT-UNet [18].

However, these procedures modify a simple aspect of the U-Net architecture, the encoder and decoder structure. A recent U-Net variation is μ-Net, which reduces the vanishing gradient phenomenon and accelerates the model convergence with weight matrices in the decoder, which are the transposes of those in the encoder. However, this strategy must be improved for complex boundary segmentation on variable biological structures due to the architecture depth [19]. Recently, a UNet-Mini architecture was created to segment small datasets. This architecture drastically reduces the number of parameters by updating only three elements [20], using element-wise skip connections (addition instead of concatenation) and reducing the number of layers and the kernel size.

In this paper, we propose Mu-Net, a reduced U-Net architecture dedicated to the segmentation of images from small bright-field BO datasets. To validate our results, we compare our architectural proposition with the current U-Net and UNet-Mini architectures using shallow methods, such as active contour or k-means, and an ablation study.

The procedure consists of training networks with images generated from an AAE using various loss and noise optimization strategies. We compare them with classic data augmentation strategies from [6,21], to highlight a tool for BO segmentations.

## 2. Materials and Methods

### 2.1. Resources

To our knowledge, only one of the six open-access organoid image datasets available in the literature concerns BOs with bright-field acquisition, which allows us to study the global development of these cultures [3]. Other imaging methods that require coloration and slicing, for instance, focus mainly on cell composition in BO developmental structures or organoid regions. The original dataset, consisting of 40 images, is from [22]. Twenty pathological and physiological cultures are captured with a bright-field microscope over 3 days. The grayscale images are 1088×1388 pixels. However, to execute our script in a reasonable time, the input images are cropped and resized to 250×250 pixels, maintaining the original proportions. Transform groups are composed by original images transformed by randomly chosen linear data augmentation strategies (flip flop, rotation and whitening). We created 40 transformed images to compare the AAE with a classic approach. The generated groups consist of 240 images in 6 loss groups optimized through an AAE [6] and 160 images in 4 noise groups optimized through an AAE from [21]. Each of the 6 loss groups is composed of 40 images generated with a binary cross entropy (BCE), a BCE with a normalization (BCE + L1), a least square (LS), a Poisson (POISSON), a Wasserstein loss function (Wass.), and a perceptual Wasserstein loss function (P.Wass.). Bright-field image acquisitions are characterized by shot noise [23], which cannot be generated with GAN loss optimizations [21]. Thus, we also test 4 noise groups composed of 40 images generated with Gaussian noise (the usual noise used in GAN, characterizing IRM acquisition [24]), speckle noise, salt and pepper noise (these two noises, which we consider interesting to test, are used to characterize other acquisitions in the biomedical field, such as laser, radar or microscopic transmission system [25]) and a shot noise injection [23]. For more details on the optimized generative methods used to synthesize images, please refer to [6] for loss update and normalization and to [21] for noise injection. Used datasets and amount of images are summarized in Table 1. We develop all scripts in Python 3.6 with an Anaconda framework containing Keras 2.3.1 and TensorFlow 2.1 and run the scripts on an Intel Core i7-9850HCPU with 2.60 GHz and an NVIDIA Quadro RTX 3000s GPU device. 

### 2.2. Segmentation Architectures

We build an architecture based on the U-Net architecture because we want to reduce execution time while conserving precision. From this perspective, we compare U-Net with UNet-Mini and our Mu-Net architecture. The differences between the three models are depicted in Table 2. 

#### 2.2.1. U-Net

Originally created in 2015 by [5], U-Net creates detailed segmentation maps that are particularly effective in the biomedical domain due to its two-part (encoder and decoder) U-shaped architecture, which spreads the contextual information of images inside the network. The encoder part employs a CNN architecture: two 3×3 3 convolutions followed by a ReLU activation unit and a max-pooling layer repeated several times. U-Net differs from previous CNNs in its second part, called the decoder part, where at each stage, the feature map is upsampled using 2×2 upconvolution. In the encoder part, the feature map from the corresponding layer is cropped and concatenated onto the upsampling map, which is a long skip connection. The final stage consists of a 1×1 convolution to reduce the feature map and produce the segmented image.

#### 2.2.2. UNet-Mini

Small dataset segmentation is a challenge in the domain. In 2020, ref. [20] proposed a new architecture based on U-Net, with a few changes: they updated the skip connections with the addition of an elementwise product instead of concatenations; they reduced the layer number to four to reduce overfitting situations; and they reduced the number of kernels used in each convolution (ks = 3×3 and stride s = 1. They used a soft-max activation task for non-binary classification.) [20]. This last update reduces the execution time and the number of hyper-parameters to learn (from 128 K instead of 17 M with the original implementation).

#### 2.2.3. Mu-Net

Our reduced U-Net architecture contains 4 similar UNet-Mini encoder–decoder layers with 16, 32, 64 and 64 kernels, as our objective is to reduce the execution time see Figure 1. However, we remove the original skip connections and instead use concatenation for feature reusability and compaction. We also update the size strides (the number of steps between filter applications to the input image) when we downsample in convolution layers, whereas the stride stays at 1 regardless of the convolution layer in UNet-Mini. We use an Adam optimizer with a binary cross entropy loss and a sigmoid activation (for binary segmentation). The model is summarized in Table 3.

### 2.3. Training

To segment the original images, we train the chosen architectures in a leave-one-out strategy summarized in Figure 2: all original and augmentation groups are used during the training stage except one original image, which is segmented during the test phase using the generated model. We process 40 leave-one-out training models × 6 loss groups + 40 leave-one-out training models × 4 noise groups + 40 leave-one-out training models using a classic augmentation dataset constituted by flip-flopping, rotating, or whitening of original images, resulting in 440 training launches for each model. We retrieve the training time of the models. To end the training and avoid overfitting, we retrieve the training time before the loss value plateaus. Original, transformed and generated images are manually segmented with ITK-SNAP software to obtain ground truth images [26].

To compare the training models, we calculate the execution time of each launch, of each leave-one-out loop, the GPU memory required to run the process, the memory required for model weights, the total number of model parameters and the number of FLOPs. To compare the number of FLOPs across models, we also calculate the FLOPs per parameter ratio.

### 2.4. Classical Segmentation Methods

We next compare the segmentation results of Mu-Net with those of classic segmentation methods, including an Otsu thresholding strategy from [27], a region growing algorithm [28], a Kmean clustering with a particle swarm optimization from [29], an active contour from [30], a level set from [31], and a watershed method from [32]. The threshold is set to 185 pixels. For the region growing algorithm, we set the regional threshold to 0.2 and the size of growth to 1. 

### 2.5. Comparison of Segmentations

To compare ground truth cerebral organoid content segmentation (GT) and derivatives of U-Net (*u*) under various conditions, mean Dice scores are calculated as
(1)$$Dice_{\left(GT,u\right)}=\frac{2|GT\cap u|}{\left|GT|+|u\right|}$$

Using the true positive TP (a pixel identified as BO content which is BO content in the ground truth GT), false positive FP (a pixel identified as BO content, which is actually background content in the GT), true negative TN (correctly identified as background content) and false negative FN (incorrectly identified as BO content) values, we calculate the accuracy, the specificity, the sensitivity, and the F1-score of the methods. The accuracy is the ratio of true to positive labels:(2)$$Accuracy=\frac{TP+TN}{TP+FP+TN+FN}$$

The sensitivity is the ratio between how many pixels were correctly identified as positives and all positive pixels:(3)$$Sensitivity=\frac{TP}{TP+FN}$$

The specificity is the ratio between how many pixels were correctly identified as negative and all negative pixels:(4)$$Specificity=\frac{TN}{TN+FP}$$

The precision is the ratio between how many pixels were correctly identified as positives and all pixels labeled as positives:(5)$$Precision=\frac{TP}{TP+FP}$$

The F1-score allows us to summarize the precision and recall (sensitivity) through a unique metric:(6)$$F1-Score=2*\frac{Precision*Sensitivity}{Precision+Sensitivity}$$

### 2.6. Visualisation

To highlight real/false positive/negative segmentation, we create a superimposed image composed of the ground truth and a sample of each segmentation resulting from the various trainings. We update the pixel values in light pink for the FP cerebral organoid segmentations and in light green for the FN.

### 2.7. Ablation study

To validate the benefits and inconveniences of our methodology, we perform an ablation study between UNet, UNet-Mini and our architecture in terms of scores. As a result, we tested N models with the following configurations:Unet,“Layer”,“filter”,“kernel”,“Ewise”,“layer + filter”,“layer + kernel”,“layer + ewise”,“Filter + kernel”,“Filter + ewise”,“Kernel + ewise”,“layer + filter + kernel”,“layer + kernel + ewise”,“layer + filter + ewise”,“filter + kernel + ewise”,“layer + filter + kernel + ewise”,“layer + filter + kernel + stride” (Mu-Net),“layer + filter + kernel + ewise + stride + activation” (UNet-Mini).

We determine a binary content and always use a sigmoid activation.

## 3. Results

We suggest completing segmentation tasks using a leave-one-out strategy (n = 79 for training and n = 1 for testing for each data augmentation strategy). We choose the classic U-Net architecture and its two lighter architectures: UNet-Mini and Mu-Net.

### 3.1. Qualitative

We assume that the data augmentation strategy used during each architecture training time could modify the segmentation results. Table 4 shows a comparison of segmented samples from U-Net, UNet-Mini and Mu-Net using the AAE loss optimizations against samples resulting from classic data augmentation strategies, while Table 5 presents segmented samples according to the AAE noise injection. As shown in Table 4 the U-Net architecture seems to be the most accurate, whereas the UNet-Mini and particularly Mu-Net segmentations exhibit high numbers of false-positive pixels under all strategies except for the Wasserstein loss optimization strategy. As shown in Table 5, U-Net and Mu-Net exhibit similar results, including a small number of false positives around the shape of the cerebral organoid. UNet-Mini results in a particular false positive around the contours of the images, and the salt and pepper noise results in a large number of false negatives.

### 3.2. Quantitative

As we cannot decipher the real precision of segmentations using human observation on a single image from a dataset, we calculate median scores for each case summarized for loss optimizations, presented in Table 6 and for noise injections, presented in Table 7. Dice scores are higher for all architectures under AAE loss optimization strategies than under classic data augmentation. The highest Dice score is achieved by the U-Net architecture when AAE Perceptual Wasserstein loss optimization is used during training. Use of the loss optimization in UNet-Mini training also results in a high Dice score, but this score is lower than that of the U-Net architecture. As shown by the median Dice scores in Table 7, Mu-Net seems to be the most accurate, followed by U-Net and UNet-Mini. Images from an AAE optimized with a Gaussian noise injection used during Mu-Net training exhibit the highest Dice scores. However, other noise injections in the generative process do not improve segmentation performance.

To summarize, U-Net exhibits the most accurate segmentation results for bright-field images of BOs. However, Mu-Net is a promising alternative that achieves almost the same accuracy as the classic data augmentation strategy, particularly in execution time. UNet-Mini would also be a solution if it achieved similar results in the noise augmentation cases. However, under the salt and pepper, the speckle or shot noise optimized synthetic images used during the training, the results of Mini-Unet are too far from the highest ranking results to consider the shape well segmented.

To verify the effect of each component of the architecture on the segmentation precision of bright-field images in a specific data augmentation strategy, we conduct an ablation study that consists of sequentially suppressing or adding an architecture component and calculating the median Dice scores for each intermediate architecture. These results are summarized in the lowest parts of Table 6 and Table 7. Initially, the layer reduction increases the Dice score regardless of the data augmentation strategy realized, whereas filtration does not improve the segmentation results. Kernel reduction results in the highest Dice score for the binary cross-entropy. Layer reduction of the U-Net architecture increases the segmentation accuracy, particularly when images generated with perceptual Wasserstein loss are used during the training of this model. Similar results are obtained with a layer, filter and kernel reduction combined with images generated with BCE loss used during the training. The best results are obtained when P.Wass loss is used for the generated images during the training time. The best noise injection results are obtained using Gaussian noise regardless of the reduction made.

### 3.3. Computational Comparisons

We verify our assumption by measuring the average execution time for training and testing a single image and for the full leave-one-out strategy (40 images) of each architecture, regardless of the data augmentation strategy used for training. These results are summarized in Table 8. As assumed, UNet-Mini and Mu-Net achieve faster executions than the original implementation: reduced architectures require 2 h for one launch and, respectively, 104 and 100 h for a leave-one-out loop; U-Net architecture requires 16.5 h for one loop and 660 h for the full leave-one-out procedure. Reduced architectures require less memory to be executed than U-Net. However, U-Net results in a higher number of FLOPs than the reduced architectures. This is possibly because the total number of processes is increased for this architecture. Indeed, if this number of FLOPs is compared to the number of parameters, Unet-Mini exhibits the highest number of FLOPs per parameter. The memory and FLOP results of Mu-Net approach the UNet-Mini efficiency. While Mu-Net achieves the best execution time, UNet-Mini also follows this trend. In summary, UNet-Mini and Mu-Net are similarly effective in terms of computational cost, and both perform better than U-Net.

### 3.4. Mu-Net and Machine Learning Comparisons

#### 3.4.1. Qualitative Analysis

A sample of each segmentation is presented in Figure 3. The threshold, active contour, Kmean and Mu-Net seem to render the best qualitative results. The region growing algorithm and threshold render the worst watershed results, possibly due to their lower precision linked to the manual setting of some parameters.

#### 3.4.2. Quantitative Analysis

The metric comparisons between the GT and each segmentation, presented in Table 9, are consistent with the previous observations in Figure 3. A comparison of the indices of the Mu-Net and classic machine learning (ML) methods shows that the best results are rendered with the Kmean algorithm. However, this algorithm does not outperform the proposed method in terms of the Dice score and specificity but rather in terms of the F1-score, sensitivity and accuracy. The other segmentation methods do not achieve high scores, with the threshold and region growing algorithms performing the worst.

## 4. Discussion

The main outcomes of this study are as follows: we present a new light segmentation methodology (Mu-Net), similar to architectures from the literature and employing various data augmentation strategies, to segment small bright-field image datasets of BOs, which have particular formations that are essential to extract. The presented framework, based on the U-Net network, runs the quickest for a single launch or for an entire leave-one-out process but has a computational efficiency similar to that of UNet-Mini. Mu-Net exhibits nearly the highest precision scores obtained with U-Net. It reaches these scores using classic data augmentation strategies during training, whereas similar scores obtained with U-Net use optimized AAE strategies. The ablation study confirms that another intermediate architecture, based upon a single layer reduction, could be used to obtain higher scores. UNet-Mini does not obtain the highest scores in some noise optimization cases, rendering this architecture less robust than Mu-Net or U-Net. Some ML segmentation algorithms seem ill-suited for extracting the shape of BOs without optimization through preprocessing, and these algorithms are only semiautomated. Mu-Net outperforms almost all the ML procedures except the Kmean clustering algorithm according to some scores.

To our knowledge, this is the first automated segmentation dedicated to a small dataset of two-dimensional bright-field images of BOs. Segmentation of small datasets (which are not always labeled) remains an issue in the biomedical image domain. Segmentation is crucial for extracting the morphological shapes and characterizing the physiological or pathological development of BOs to form a brain in vitro model that could fill the gaps of in vivo models [1]. Only U-Net has been used for automatically segmenting ventricular zones in the Scout software for clarified BO images and for extracting the shapes of two dimensional BO images to validate the accuracy of the data augmentation strategy [4,6]. However, the training time, the required number of labeled images and the approximate segmentation of neuro-epithelial regions (essential for determining whether a culture is well developed) convinced us to find an appropriate solution for these particular cultures.

Among light U-Net architectures, Mu-Net and UNet-Mini reduce the training time. This could be explained by the drastic reduction of learned hyperparameters and the layer reduction in these networks [16,20]. Usually, the architecture is specified to produce highly accurate segmentation. However, this involves higher processing and computation costs [13,14,15]. Unlike U-Net derivatives applied to other biomedical images (3D, attention, inception, residual, current neural network, and adversarial), the reduced architecture requires few updates [7] and could be quickly implemented on image segmentation platforms.

Mu-Net requires a small number of parameters, processing memory, and higher FLOPs/parameters than U-Net; however, it does not achieve better computational results than UNet- Mini. This reduced performance improves resource consumption with respect to the number of models to process, benefiting the small architecture field. The model performance needs to be tested in other domains. A Mu-Net optimization could be useful for achieving computation performances similar to those of UNet-Mini. However, Mu-Net exhibits higher segmentation quality than UNet-Mini. Additionally, UNet-Mini is less robust than Mu-Net for maintaining high Dice scores regardless of the noise optimization used in the training step. Thus, Mu-Net is, in our opinion, the best alternative to U-Net, which requires more resources than UNet-Mini. Future research could concern the best compromise with respect to computational performance, specifically the number of parameters, to perform qualitative segmentation that is satisfactory for biomedical experts. Indeed, reduced architectures run faster and require less consumption and resources while achieving segmentation levels that allow shape characterization. In our opinion, architecture reduction is beneficial only if it does not degrade the segmentation quality. Mu-Net may be limited from reaching similar high results on other datasets with higher resolutions. However, to overcome this potential issue, employing an attention module in our reduced architecture can improve the result [13,14].

Mu-Net is almost as accurate as U-Net depending on the data augmentation strategy used during training. In contrast, UNet-Mini never achieves higher Dice scores than 0.85, despite being implemented similarly to Mu-Net. This is explained by the transfer learning method used during training: while U-Net and our Mu-Net architecture use concatenations for long skip connections, UNet-Mini uses an additional long skip connection based upon an elementwise product [5,20]. A skip connection skips layers in the neural network and uses the output of one layer as the input to future layers. In addition, data captured in the initial layers are features corresponding to lower semantic information that are extracted from the input and shared to the output. In concatenation, low-level information is shared between the input and output, and this information is directly shared across the network. This method allows feature compaction and high standardization, which is not possible using addition.

Mu-Net achieves its highest Dice scores when a classic data augmentation strategy is used during its training. The ablation study confirmed that combining layer reduction and kernel size maintains high Dice scores in segmentation, whereas using only layer reduction does not. We propose combining the reduction layers and kernel size and hypothesize that concatenation allows transfer learning on essential features. The classic data augmentation chosen here is supported by simple transformations such as reflections, rotations of 90 degrees or filtered whitening of images. GAN solutions are based on generative and discriminative networks, which have to be trained and optimized. If this behavior is observed with another dataset of bright-field images, this architecture could become more attractive (for avoiding using data augmentation strategies based upon GAN architectures, which is time consuming).

A high rate of false positives is present in the samples chosen to illustrate the architecture results under classic data augmentation. A false positive is defined as an instance of an organoid region that is present in the predicted mask and has zero intersection with the ground truth mask region. The surroundings of BOs include spreading cells [2] that could be interpreted by these networks as organoid regions. The ground truth could impact this result due to the small dataset used for training and the more shallow architecture, which could cause surrounding details to be approximately learned. Observations of the segmentation results using all data augmentation strategies in the same training dataset could be interesting.

Other segmentation strategies do not achieve strong results. The main drawback observed in this study is the need to manually preset some parameters, rendering the segmentation process semiautomated. These unexpected scores are also linked with the fact that ML algorithms are inappropriate for datasets with strong variation [33]. However, in the early developmental stage of BOs, no neuroepithelial formation is exhibited, whereas after 14 days, the BO shape is transformed by this structure [22]. The algorithms are also influenced by batch syndrome: BOs grow differently in the same environment, resulting in one or more neuroepithelial zones [1]. Thus, instead of of furnishing a preset parameter for each image in a particular dataset, ML solutions do not consider these cultures.

The Kmean machine learning algorithm is an exception, achieving the best segmentation scores. By separating the number of parts to segment (here, the background and the BO), this algorithm does not require another setting that influences the score. The Kmean clustering algorithm uniquely compares the cluster containing a centroid with a pixel [29]. Particle swarm optimization allows us to obtain a more precise clustering efficiency and thus a better accuracy than using a single Kmean algorithm.

Mu-Net outperforms all ML strategies except Kmean. Our reduced model of segmentation for a tiny dataset of BO images is automated and does not require presetting parameters. The light architecture allows the model to perform image segmentation almost as quickly as using a Kmean algorithm, producing similar results. However, this architecture currently needs to be improved for segmenting tiny BO datasets. An improvement could be to use prior information during Mu-Net training (such as providing shape or topological information on the kind of image segmented) [34], but this requires a better understanding of the morphological characterization of these cultures.

The proposed Mu-Net architecture could also be compared with other U-Net variations, such as those described in [7]. We observe the fundamental role of concatenation in this architecture; however, other U-Net variations are characterized by dense connections [35]. Combining such an update with the lighter architecture to observe its contributions to the bright-field image segmentation of these cultures could be interesting. Additionally, we choose a supervised version (with manually labeled datasets as ground truth in this paper. In future comparisons, the identification of a zero-shot learning solution to decrease the time/labor of neurobiologists in manual segmentation could be interesting [36]. Concerning the metrics comparison, we could calculate other known scores (F1, accuracy, etc.) in future studies. Augmentation strategies could also be formed with nonlinear transformations or other GAN architectures [37], and other data augmentation strategies could be tested with semantically optimized segmentation using an optimized U-Net [38]. We mention that the noise injection optimization benefit during training must be thorough. We verify the contributions of the ablation study, which includes only a few configurations. In the future, conducting an ablation study on filtering, kernel reduction, or even other skip connections would be interesting.

The main future objective is to reproduce these procedures on another bright-field dataset of BO images or on another biomedical culture (miBrain, cortical, etc.) if no BO dataset is available. Concerning the dataset we use [22], a quantitative comparison of the segmentation quality at various stages (9, 14 and 15 days of development) and states (physiological and pathological) could be interesting. In this paper, we quantify the segmentation quality only for the entire dataset, proving that Mu-Net is robust for all states and stages. Once all the procedures are validated on other datasets, these reduced algorithms could also be used in biosensing, as described in [39]. Indeed, Mu-Net could be used for real-time, label-free detection of biological components in a cost-effective way.

## 5. Conclusions

In this article, we report the first results on Mu-Net, a light deep architecture for bright-field BO image segmentation in small datasets. The proposed segmentation architecture achieves state-of-the-art performance (U-Net) with a faster computation time than other architectures. Mu-Net also outperforms other computer vision segmentation algorithms and is particularly robust to various data augmentation strategies, unlike UNet-Mini. To consolidate these initial results beyond validating our methodology on other datasets, we also need to examine the particular contributions of combining a lightened architecture with other SOTA methodologies. We believe that optimization, including that of shape/topological information given during the data augmentation step to better train our model, could improve the segmentation.

## Figures and Tables

**Figure 1 biomedicines-11-02687-f001:**
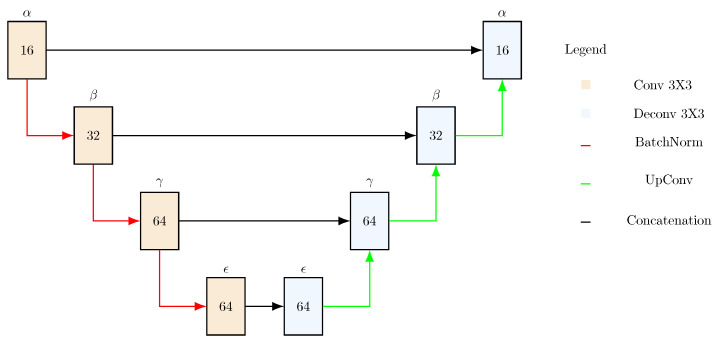
Mu-Net architecture. Arrows represent operations, and blocks represent convolutional or deconvolutional layers constituted by two 3×3 convolutions (except for the latest blue block, which is constituted by two 3×3 and one 1×1 deconvolution). The input consists of 250×250 pixel images.

**Figure 2 biomedicines-11-02687-f002:**
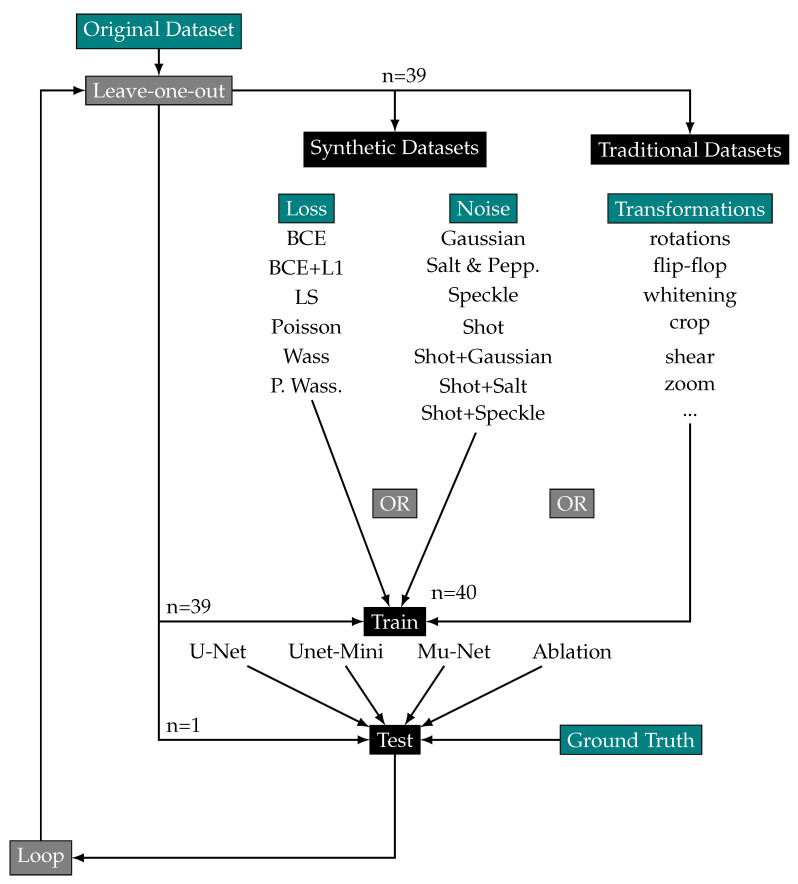
Experimental scheme of the leave-one-out strategy to test the impact of various data augmentation methods on segmentation. The synthetic and traditional datasets are generated at each loop without the leave-one-out image.

**Figure 3 biomedicines-11-02687-f003:**
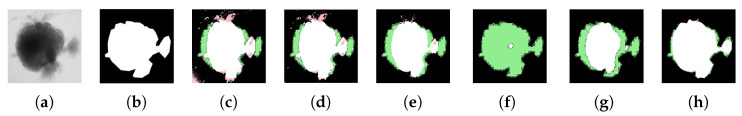
Qualitative comparisons of segmentations methods (**a**) Original image from [22]; (**b**) Ground truth; (**c**) Threshold; (**d**) Active contour; (**e**) Kmeans; (**f**) Region; (**g**) Watershed; (**h**) Mu-Net.

**Table 1 biomedicines-11-02687-t001:** Sample view for each dataset. (1) is from [22], (3) and (4) samples are respectfully from [6,21]. Only samples from (2) are purposely transformed from the original dataset (1).

Dataset	Effectives	Sample
(1) Original	40	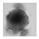
(2) Transformed	40	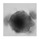
(3) Loss optimization AAE	240	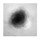
(4) Noise injection AAE	160	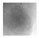

**Table 2 biomedicines-11-02687-t002:** Summary of differences between U-Net and derivatives architectures.

Update Name	Unet	UNet-Mini	Mu-Net
Layer Number	5	4	4
Filter	[64, 128, 256, 512, 1024]	[16, 32, 64, 64]	[16, 32, 64, 64]
Stride	default (1)	default (1)	1 or 2
Activation	Sigmoid	Softmax	Sigmoid
Skip connections	Concatenation	Addition	Concatenation

**Table 3 biomedicines-11-02687-t003:** Mu-Net Model. ‘k’ corresponds to the kernel size, ‘s’ to the stride, ‘a’ to activation, ‘d’ to dropout, ‘p’ to pool size, ‘sz’ to upsampling size. Input is images of 250×250 pixels.

Name	Filter	Parameters
Conv	16	[k = 3, s = 1, a = relu]
Conv	16	[k = 3, s = 2, a = relu]
BatchNorm		
Conv	32	[k = 3, s = 1, a = relu]
Conv	32	[k = 3, s = 2, a = relu]
BatchNorm		
Conv	64	[k = 3, s = 1, a = relu]
Conv	64	[k = 3, s = 2, a = relu]
BatchNorm		
Conv	64	[k = 3, s = 1, a = relu]
Conv	64	[k = 3, s = 2, a = relu]
BatchNorm		
Dropout		[d = 0.5]
Maxpooling		[p = 2.2]
Deconv	64	[k = 3, s = 1, a = relu]
Deconv	64	[k = 3, s = 2, a = relu]
Upsampling	64	[k = 2, a = relu, sz = 2.2]
Concatenation		
Deconv	64	[k = 3, a = relu]
Deconv	64	[k = 3, a = relu]
Upsampling	32	[k = 2, a = relu, sz = 2.2]
Concatenation		
Deconv	32	[k = 3, a = relu]
Deconv	32	[k = 3, a = relu]
Upsampling	16	[k = 2, a = relu, sz = 2.2]
Concatenation		
Deconv	16	[k = 3, a = relu]
Deconv	16	[k = 3, a = relu]
Conv2D	2	[k = 3, a = relu]
Conv2D	1	[k = 1, a = sigmoid]

**Table 4 biomedicines-11-02687-t004:** Observation of segmentation achieved with several noise optimization data augmentation strategies while training U-Net and its derivative architectures. GT corresponds to the ground truth segmentation. False-positive pixels are colored in purple, false negatives in cyan, true negatives in black and true positives in white.

	GT	Classic	Gaussian	Salt & Pepp.	Speckle	Shot
U-Net	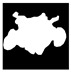	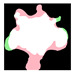	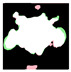	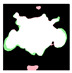	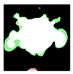	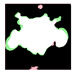
Unet-Mini	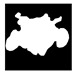	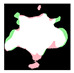	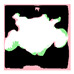	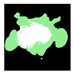	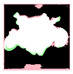	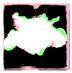
Mu-Net	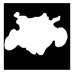	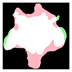	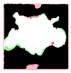	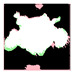	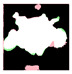	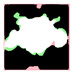
Legend	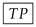	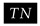	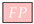	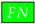		

**Table 5 biomedicines-11-02687-t005:** Observation of segmentations achieved with several loss injections for the data augmentation strategy while training U-Net and its derivative architectures. GT corresponds to the ground truth segmentation. False-positive pixels are colored in purple, false negatives in cyan, true negatives in black and true positives in white.

	GT	Classic	BCE	BCE + L1	LS	Poisson	Wass.	P.Wass.
U-Net	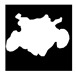	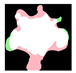	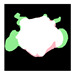	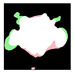	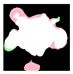	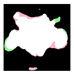	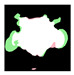	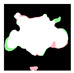
Unet-Mini	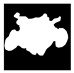	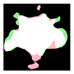	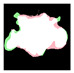	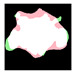	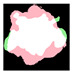	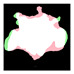	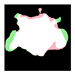	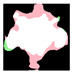
Mu-Net	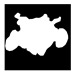	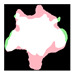	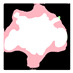	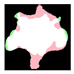	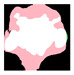	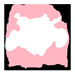	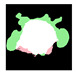	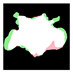

**Table 6 biomedicines-11-02687-t006:** Dice scores for segmentation with several loss optimizations of data augmentation strategies during training. The best results per row are bolded.

	Classic	BCE	BCE + L1	LS	Poisson	Wass.	P. Wass.
**U-Net**	0.68	0.87	0.87	0.86	0.87	0.88	**0.90**
Layer	0.83	0.89	0.88	0.87	0.90	0.88	**0.91**
Filter	0.85	0.56	0.84	0.82	0.83	**0.85**	0.83
Kernel	**0.85**	0.17	0.84	**0.85**	0.68	**0.85**	0.64
Ewise	0.51	0.69	0.72	**0.80**	0.66	0.68	0.73
Layer + Filter	0.84	0.87	0.87	0.85	**0.88**	0.87	**0.88**
Layer + Kernel	0.34	0.35	0.68	0.68	0.35	0.50	**0.70**
Layer + Ewise	0.33	0.21	0.28	0.28	0.25	0.32	0.37
Filter + Kernel	0.71	**0.78**	0.72	0.59	0.70	0.71	0.74
Filter + Ewise	0.67	0.39	**0.77**	0.76	0.59	0.31	0.65
Kernel + Ewise	0.17	**0.65**	0.55	0.59	0.64	**0.65**	0.56
Layer + Filter + Kernel	0.83	**0.91**	0.88	0.86	0.87	0.88	0.87
Layer + Kernel + Ewise	0.35	0.33	0.16	0.16	0.21	0.31	0.37
Layer + Filter + Ewise	0.42	0.28	0.19	0.36	0.13	0.29	0.41
Filter + Kernel + Ewise	0.50	0.34	0.29	0.29	0.45	0.37	0.21
Layer + Filter + Kernel + Ewise **(Mu-Net)**	0.85	0.88	0.87	**0.89**	0.87	**0.89**	0.88
Layer + Filter + Kernel + Stride	0.84	0.83	**0.85**	0.78	0.84	0.82	0.81
Layer + Filter + Kernel + Stride + Activation + Ewise **(Mini-UNet)**	0.86	0.84	0.87	**0.88**	0.85	**0.88**	0.82

**Table 7 biomedicines-11-02687-t007:** Dice scores for segmentation with several noise optimization data augmentation strategies during training. The best results per row are bolded.

	Classic	Gaussian	Salt & Pepp.	Speckle	Shot
**U-Net**	0.68	**0.83**	0.82	0.74	0.74
Layer	0.83	**0.91**	0.83	0.83	0.85
Filter	0.85	**0.83**	0.52	0.73	0.74
Kernel	0.85	0.64	0.72	0.58	**0.75**
Ewise	0.51	**0.73**	0.22	0.18	0.38
Layer + Filter	0.84	**0.88**	0.83	0.69	0.80
Layer + Kernel	0.34	0.70	**0.77**	0.49	0.40
Layer + Ewise	0.33	0.20	0.54	0.20	0.32
Filter + Kernel	0.71	0.74	0.59	0.55	0.62
Filter + Ewise	0.67	**0.65**	0.45	0.10	0.21
Kernel + Ewise	0.17	0.56	0.45	0.11	0.26
Layer + Filter + Kernel	0.83	**0.87**	0.81	0.77	0.74
Layer + Kernel + Ewise	0.35	0.42	0.37	0.26	0.32
Layer + Filter + Ewise	0.42	0.18	0.14	0.08	0.14
Filter + Kernel + Ewise	0.50	0.21	0.29	0.26	0.16
Layer + Filter + Kernel + Ewise **(Mu-Net)**	0.85	**0.88**	0.80	0.81	0.83
Layer + Filter + Kernel + Stride	0.84	0.81	0.59	0.75	0.81
Layer + Filter + Kernel + Stride + Activation + Ewise **(Mini-UNet)**	0.86	**0.82**	0.74	0.48	0.77

**Table 8 biomedicines-11-02687-t008:** Comparison of training time, memory usage, model parameters and FLOPs per segmentation architecture. Training time is given for both a single image (1 launch) and the entire leave-one-out process (LOO on 40 images).

Architecture	Time 1 Launch (s)	Time LOO (s)	Memory GPU (GB)	Memory Model (MB)	Model Param.	Flops (MFLOP)	Flops/ Param.
U-Net [5]	990	39,613	35	118	17 M	1092	9
UNet-Mini [20]	157	6282	7.35	1.31	128 k	40	316
Mu-Net (ours)	151	6044	7.83	2.60	680 k	45	58

**Table 9 biomedicines-11-02687-t009:** Comparison of segmentation methods.

	Threshold	Active	Kmean	Region	Watershed	Mu-Net
Dice	<0.1	0.62	0.89	<0.1	0.61	0.88
F1-score	0.40	0.66	0.72	0.20	0.61	0.77
Sensitivity	0.39	0.69	0.77	0.19	0.53	0.90
Specificity	0.96	0.84	0.89	0.86	0.94	0.86
Accuracy	0.83	0.90	0.90	0.7	0.88	0.92

## Data Availability

Data are available upon corresponding author request.

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
