# Peer review of "Mu-Net a Light Architecture for Small Dataset Segmentation of Brain Organoid Bright-Field Images"

_biomedicines, 2023, doi:10.3390/biomedicines11102687_

Round 1
Reviewer 1 Report
The manuscript described a new machine-learning method (Mu-Net) for small dataset segmentation of brain organoid bright-field images, as an improvement to the previously established U-Net method. The authors compared several data augmentation strategies and concluded that a U-Net segmentation trained on optimized augmentation produces the best result. However, the authors claimed that the Mu-Net has several easier-to-use features with no worse outcome.
The abundant technical details in the manuscript appear to support the authors’ conclusion. However, the criteria of brain organoid segmentation used here appears to be rather crude. Essentially, the imaging processing detected the border of an organoid, not the internal ventricular zones. Detection of the latter especially with bright-field images would be the desired outcome. Simply detecting the tissue border is not convincing enough to support the new Mu-Net method, regardless of its usability.
The authors claimed “The originality of our current work lies in the fact that we train networks with images generated from an AAE using various loss and noise optimizations and, compared with classical data augmentation strategies from [6,18], to highlight a tool dedicated to BO segmentations” – This sentence seems to describe the procedure and fails to point out the “originality”.
Further development of this method is needed to demonstrate successful segmentation of incongruous internal structures of a brain organoid.
Typos throughout the manuscript, just a few examples here:
1. “…modelate” ?
2. “that are often non automatic and non adapted to their non spherical shape”
3. “created by [5]” – please spell out the group, not usual to just put a reference number as an incomplete sentence.
4. “the expanding path enables to detect” – Change to “enables the detection of”
5. “[10] reduce only convolutional…” – again, not usual to just put a reference number as an incomplete sentence.
6. “The difference between U-Net and others previous CNN…” --? Change to “other previous…”
7. “We test an Otsu thresholding from [22],…” -- again, not usual to just put a reference number as an incomplete sentence.
8. “Table 4 represents segmented samples from U-Net, UNet-Mini and Mu-Net according to the AAE loss optimisations against classical data augmentation strategies while Table 4 represents..”—should be Table 5?
9. “if the culture is well developed) convinced us to found an appropriate solution..” – change to “find”.
10. “Set most of the time manually, and despite the fact of testing various combinations, these parameters render the segmentation process semi-automated and, especially not expected scores.” – jumbled sentence, hard to understand.
11. “Taking appart the number of parts to segment” – change to “apart”
12. “ of another biomedical model if there is no available” -- ?
Author Response
We thank the Reviewer1 for the comments and typos and we would like to answer the comments below.
First we would like to precise our aim is to automatically extract brain organoid shapes in order to characterize their global development and not a particular region.
This information could help to decide if a culture is growing or not in the culture medium.
Ventricular zone characterisation and segmentation of the incongruous internal structures of brain organoids is a future step after the global shape segmentation and characterisation have been validated.
We corrected the given typos and updated the manuscript.

Reviewer 2 Report
The manuscript ID biomedicines-2450063 has been devoted to mainly study about light U-segmentation. The authors report the reduction of the training time while increasing the sensitivity with small input datasets. Results related to brain organoids bright-field images were analyzed. Please see below some points to the authors:
1. Please comment how was selected the size of the original data set and the images studied for this study.
2. The authors wrote “to execute our script in a reasonable time, the input images are cropped and resized to 250 × 250 pixels, maintaining the original proportions.” Please comment about the time related to the processing and how much can be reduced the images for maintaining the original proportions.
3. If possible, please comment with better details how can be identified or analyzed the noise in the images studied as described in figure 3.
4. Influence of external factor during the acquisition of the images over the 3 days mentioned before processing could be described.
5. Is there a preference to study bright field images instead of other techniques for the research of brain organoids by this technique?
6. Some perspectives about the potential use of this technique in other areas for biosensing assisted by machine learning could be envisioned. The authors are invited to see for instance: https://doi.org/10.3390/bios12090710
7. In the discussion section, the authors should state what this work adds to literature in respect to updated publications in the topic. You can see for instance https://doi.org/10.3390/rs15071838
8. Only one reference 2023 is cited as a reference. The bibliography could be updated.
9. The advantages and disadvantages of the proposed architecture must be highlighted in the conclusion section.
10. The parameters in the equations must be declared for a general reader.
A proofreading is suggested.
Author Response
We thank the Reviewer2 for the clarity recommendations and answer to the questions here.
- We reduce the size of the images in order to use the same size as in the literature using deep learning architectures to segment images.
- The execution time of a process with non reduced images requires at least 4 times the execution time we mention in the manuscript and requires to execute the manuscript on a virtual machine more powerful than the computer cited in resources. We reduce the size until reaching the literature standard and maintain proportions by cropping and then reducing the size due to our objective to retrieve the exact shape (see ref 6).
- Noise in images has been calculated with a Laplacian Matrix and then the pixel variance is calculated. Then we calculate the median noise in all original dataset which aim at giving a level of noise to reach with the generative method . Synthetic shot noise peak for instance is then increased until reaching this median seen as the threshold to obtain a similar noise as given in original images (see ref 19).
- The shot noise is a particularity of bright-field acquisition and due to the quantum nature of electrons, and it can be modeled using the Poisson distribution. For more precisions see our previous contribution on Noise optimizations for AAe (ref 19). Images are acquired over three days but are not concerning the same brain organoids growing.
- Bright-field acquisition allows to follow the development of these cultures without a specific preparation (slicing, clarifying or coloring the culture). In the literature this acquisition constitutes 5% of the mentioned acquisition, but most of the time is not mentioned and is the first step to observe brain organoid in their culture (see ref 3).
- This architecture may be used in other areas such as biosensing, however, it has to be tested first and could be​​ a perspective.
- We added some related recent references in the field in introduction and discussion.
- Advantages and disadvantages have been added in conclusion.
- Parameters are declared above the equations.

Reviewer 3 Report
1. In the model introduction, there is no specific formula to provide detailed information about the model.
2. In order to improve the reliability of the experiment, is it necessary to change the composition of the dataset, the proportion of loss groups and noise groups, and conduct multiple experiments?
3. It is best to indicate the size of the image in Table 2 and Figure 2. The naming of the decoder stage in the table is inconsistent with the one in the figure. The explanation is too brief. DConv2D did not say what it is, does the downsampling multiple not match the upsampling, and why is the number of filter channels 16 × 16. Shouldn't it be one-dimensional?
4. In Table 3, to illustrate the lightweight of the model, Flops and Paras should be indicated. This article has repeatedly mentioned reducing execution time, preferably by adding the time required to run an epoch.
5. It is best to add a legend to Table 4 to make it clearer. The layout of the table should not be inserted between the introduction of the ablation model.
6. There is no significant improvement in the comparison between the model in this article and the results of MINI UNet.
7. The conclusion section does not highlight the lightweight and speed block characteristics of the model in this article.
8. The paper does not compare some existing deep learning methods, including some lightweight methods. There are currently many lightweight methods for deep learning.
1. In the model introduction, there is no specific formula to provide detailed information about the model.
2. In order to improve the reliability of the experiment, is it necessary to change the composition of the dataset, the proportion of loss groups and noise groups, and conduct multiple experiments?
3. It is best to indicate the size of the image in Table 2 and Figure 2. The naming of the decoder stage in the table is inconsistent with the one in the figure. The explanation is too brief. DConv2D did not say what it is, does the downsampling multiple not match the upsampling, and why is the number of filter channels 16 × 16. Shouldn't it be one-dimensional?
4. In Table 3, to illustrate the lightweight of the model, Flops and Paras should be indicated. This article has repeatedly mentioned reducing execution time, preferably by adding the time required to run an epoch.
5. It is best to add a legend to Table 4 to make it clearer. The layout of the table should not be inserted between the introduction of the ablation model.
6. There is no significant improvement in the comparison between the model in this article and the results of MINI UNet.
7. The conclusion section does not highlight the lightweight and speed block characteristics of the model in this article.
8. The paper does not compare some existing deep learning methods, including some lightweight methods. There are currently many lightweight methods for deep learning.
Author Response
We thank the Reviewer3 for the clarity recommendations and answer to the questions here.
- All the information about the model is written in Table2.
- Updating the composition of the training dataset allows estimating which kind of training helps the segmentation to be more accurate. Trainings are compared with results from a training with simple transformations to render it reliable.
- The size has been added in figure and table 2. Names and filter size has been updated between table and figure 2 and the filter size. Downsampling and upsampling match together.
- Parameters, Flops and Memory requirements have been added in Table 8 in addition to the execution time, and we added methods, results and discussions sections related.
- We add a legend in Table 4 and corrected the juxtaposition of the model list and this table.
- The improvements between Mu-Net and Mini-Unet are in the result comparison table of noise optimizations for the data augmentation. Indeed in this table we observe Mu-Net keeps consistent results while Mini-Unet renders weak dice results especially concerning the speckle noise. Some differences have also been highlighted in new parts concerning the 4th answer.
- We updated the conclusion.
- It includes only Unet-Mini, in the future we would like to add other models comparisons such as described in the introduction section with for instance attention modules.

Round 2
Reviewer 1 Report
The Discussion & Conclusion can be more clear and succinct. For example, the sentence "Thus from an overall point of view, Unet-Mini reaches 272 the highest computation scores, except for the execution time which is slightly better for Mu-Net" (line 271-274), are the authors claiming Unet-Mini overall is better? Though the entire study is proposing Mu-Net as a better solution.
If lower computational cost is the top criteria ,then say it clearly why Mu-Net is the chosen method based on the comparison studies. If other considerations like retaining resolution/details perhaps for future regional segmentation (isn't that the ultimate goal), perhaps Mu-net is still limited.
It is still easy to get lost in the woods with all the technical details. I highly recommend the authors to clarify their conclusion, in simpler sentences without too many extraneous details.
As a reviewer from biomedical but not computational background, the significance and the impact of this work still need to stand out from the technical details.
Author Response
We thank the reviewer 1 for the useful comments and raised lacks and we replied below.
All updated information for these questions are highlighted in purple (and some sentences have been erased).
- « The Discussion & Conclusion can be more clear and succinct » We updated some lasting sentences in discussion and in all the conclusion.
- « Thus from an overall point of view, Unet-Mini reaches 272 the highest computation scores, except for the execution time which is slightly better for Mu-Net" We updated the paragraph containing this sentence. We explain at lines 244, 305, 338, 340 and 441 why Mu-Net is more robust than Unet-Mini in term of training strategies (due to the high dice scores whatever the noise optimisation used in the training step of the segmentation).
- All the updated sentences for the claim of « similar efficiency in term of time, memory and Flops for Mu-Net and Unes-Mini » are in lines 275, 300 and 324.
- Concerning the limits of our architecture for higher resolution or details level of segmentation, we mentioned it in line 433 and we propose a potential solution.

Reviewer 2 Report
In my opinion, the reviewed version of the manuscript still contains most of the issues raised in the initial review stage. The clarification of the issues that were pointed out lacks of attractive details.
Moreover, I agree with the need of original aspects indicated by reviewer #1 for the proposed methodology. Proper critical discussion and perspectives of the analytical performance correspond to characteristics that are crucial for the journal, and no substantial changes in the manuscript have been added to address this point.
A proofreading is suggested
Author Response
We thank the reviewer2 for these comments and updated the manuscript with cyan color text to clarify the pointed updates.
The dataset and size are commented at line 82.
The size selection is described at line 88.
For noise explanation and selections we updated the paragraph between line 99 and 108.
Bright-field use and explanation in this article is mentioned at the end of sentence 82-84.
We added the biosensing application since line 418.
The comparisons of this procedure with the literature are described in introduction since line 41 to 71 and discussed in paragraph 369 and paragraph 395, with the compared algorithm since 315.
In introduction and discussion 2023 references has been added in blue parts.
Advantages and disadvantages of the proposed architecture are highlighted in conclusion in blue parts.
Parameters before equations has been detailed since line 173.
We corrected also typos.
Performances which are required in this journal have been added in Table 8, described in 3.3 and discussed in paragraphs since line 315 and since line 323 in blue parts (see our answer to reviewer1).

Reviewer 3 Report
The author answered my question and I have no further questions.
Author Response
We thank again the reviewer3 for all the previous comments and questions.

Round 3
Reviewer 2 Report
The authors have provided an interesting manuscript with original ideas for segmentation of brain organoid bright-field images. The results are attractive and the conclusions are solid. In my opinion, this work can be considered for publication in present form.
Author Response
We thank the reviewer2 for the reply, and useful comments.
